# Mitigating the Functional Deficit after Neurotoxic Motoneuronal Loss by an Inhibitor of Mitochondrial Fission

**DOI:** 10.3390/ijms25137059

**Published:** 2024-06-27

**Authors:** Maria Ciuro, Maria Sangiorgio, Valeria Cacciato, Giuliano Cantone, Carlo Fichera, Lucia Salvatorelli, Gaetano Magro, Giampiero Leanza, Michele Vecchio, Maria Stella Valle, Rosario Gulino

**Affiliations:** 1Department of Biomedical and Biotechnological Sciences, University of Catania, 95123 Catania, Italy; ciuromaria@hotmail.it (M.C.); ma.sangiorgio@outlook.it (M.S.); valeria.cacciato@gmail.com (V.C.); cantonegiuliano@hotmail.com (G.C.); carl.fichera@gmail.com (C.F.); michele.vecchio@unict.it (M.V.); m.valle@unict.it (M.S.V.); 2Department of Medical and Surgical Sciences and Advanced Technologies “G.F. Ingrassia”, Anatomic Pathology, University of Catania, 95123 Catania, Italy; lucia.salvatorelli@unict.it (L.S.); g.magro@unict.it (G.M.); 3Department of Drug and Health Sciences, University of Catania, 95125 Catania, Italy; gpleanza@unict.it

**Keywords:** behavioral test, cholera toxin-B saporin, gastrocnemius muscle, Mdivi-1, mouse, neurodegeneration, spinal cord, synaptic plasticity

## Abstract

Amyotrophic lateral sclerosis (ALS) is an extremely complex neurodegenerative disease involving different cell types, but motoneuronal loss represents its main pathological feature. Moreover, compensatory plastic changes taking place in parallel to neurodegeneration are likely to affect the timing of ALS onset and progression and, interestingly, they might represent a promising target for disease-modifying treatments. Therefore, a simplified animal model mimicking motoneuronal loss without the other pathological aspects of ALS has been established by means of intramuscular injection of cholera toxin-B saporin (CTB-Sap), which is a targeted neurotoxin able to kill motoneurons by retrograde suicide transport. Previous studies employing the mouse CTB-Sap model have proven that spontaneous motor recovery is possible after a subtotal removal of a spinal motoneuronal pool. Although these kinds of plastic changes are not enough to counteract the functional effects of the progressive motoneuron degeneration, it would nevertheless represent a promising target for treatments aiming to postpone ALS onset and/or delay disease progression. Herein, the mouse CTB-Sap model has been used to test the efficacy of mitochondrial division inhibitor 1 (Mdivi-1) as a tool to counteract the CTB-Sap toxicity and/or to promote neuroplasticity. The homeostasis of mitochondrial fission/fusion dynamics is indeed important for cell integrity, and it could be affected during neurodegeneration. Lesioned mice were treated with Mdivi-1 and then examined by a series of behavioral test and histological analyses. The results have shown that the drug may be capable of reducing functional deficits after the lesion and promoting synaptic plasticity and neuroprotection, thus representing a putative translational approach for motoneuron disorders.

## 1. Introduction

Amyotrophic lateral sclerosis (ALS) is an adult-onset neurodegenerative disease characterized by gradual muscle paralysis caused by the degeneration of spinal motor neurons (MNs) and cortical neurons in motor areas. The disease progression is rapid, with a fatal outcome due to respiratory failure within 2 to 5 years after the onset of symptoms. ALS is a rare but severe disease with an incidence of approximately two people per 100,000. While >90% of ALS patients develop a sporadic form of ALS (sALS), just 5–10% of patients have a family history of disease (fALS) associated with known genetic mutations [1,2,3,4,5,6]. Anyway, both sALS and fALS show similar clinical phenotypes, suggesting that different initiating molecular insults can elicit common pathogenic pathways that lead to neurodegeneration through a multistep etiopathogenic mechanism [7,8,9,10].

The death of MNs is one of the main pathological hallmarks of ALS, and several studies have found a focal initiation of MN degeneration, probably where a series of pathogenetic factors converge to create a toxic microenvironment [9,11]. Moreover, the involvement of muscle cells and axon terminals in causing retrograde MN degeneration has been included in the pathogenic mechanism [12,13], so muscle denervation may appear during the early stages of ALS pathogenesis, and it can be observed by electromyography (EMG) [14]. In our view, the use of a simplified in vivo model of MN degeneration would help in the step-by-step dissection of ALS pathogenesis. In particular, the focal removal of confined pools of spinal MNs by intramuscular injection of cholera toxin-B conjugated to saporin (CTB-Sap) and the following retrograde suicide transport [15] has proven to be useful in mimicking dysphagia [16,17], respiratory dysfunction [18,19] and other effects of focal MN loss [20,21,22,23,24,25], and it represents a valuable tool to study compensatory plastic changes, including synaptic plasticity, axonal sprouting, and other morphological and functional adaptations [26]. These events may also take place in ALS models, where MN degeneration occurs progressively and the remaining cells probably try to compensate for the motor deficits through plastic events that could be similar to those previously described. When the progressive loss of MNs exceeds the compensatory capacity of the surviving cells, the first signs of the disease appear [27,28,29]. Despite intensive research, it remains poorly understood why MNs are specifically targeted in ALS. Among the pathogenic factors, mitochondrial dysfunction may exert a key contribution [30,31,32,33]. Several evidences have associated abnormal mitochondrial dynamics with excessive mitochondrial fission predominantly mediated by the hyperactivation of the dynamin-related protein 1 (Drp-1), a cytosolic GTPase, recruited to the outer mitochondrial membrane, where it assembles into a ring-like structure around the mitochondria, causing constriction and subsequent division [34,35]. Mitochondria are organelles that undergo rapid and opposing processes of fission and fusion, two mechanisms normally balanced to maintain a healthy mitochondrial network. It has been reported that high levels of Drp-1 trigger mitochondrial damage causing insufficient ATP production, with multiple detrimental neuronal consequences that ultimately promote apoptosis [36]. Although the impact of impaired mitochondrial dynamics on MN degeneration is still unclear, several studies suggest that the manipulation of mitochondrial fission and fusion has considerable potential for treating several neurodegenerative diseases [35].

Mdivi-1, a cell-permeable quinazolinone, is a selective pharmacological inhibitor of Drp-1, capable of inhibiting the fission process by directly decreasing the GTPase enzymatic activity of Drp-1, thus resulting in neuroprotection in models of Alzheimer’s disease, Parkinson’s disease, and multiple sclerosis [37,38]. In an attempt to determine the therapeutic impact of Mdivi-1 after MN loss, we used the already established mouse CTB-Sap model [22,25]. Given the previously observed up-regulation of Drp-1 in this model, together with increased mitochondrial fission [25] at least in the affected muscle, the present research work has investigated whether the administration of Mdivi-1 could be neuroprotective on damaged or stressed MNs, and whether it may promote spinal cord (SC) plasticity and compensatory functional restoration, as well as muscle-repairing processes.

## 2. Results

Animals were lesioned by intramuscular injection of CTB-Sap and then treated weekly with Mdivi-1 or with drug vehicle alone (DMSO) for four weeks (i.e., at 3, 10, 17, 24 days post-lesion; dpl). The first signs of motor deficit appeared as soon as two or three dpl. To evaluate the effects of the neurotoxin and drug treatments, all mice were subjected to a series of motor behavior tests (please see below). A schematic description of the experimental timeline is reported in Figure 1. Untreated control mice and animals receiving unconjugated saporin were found to be similar in all parameters, and they were pooled together in a single control group (CTRL). Their body weight was measured on a weekly basis to evaluate their general health condition (Figure 2A): as expected, CTRL mice showed a constant increase in body weight along the experimental timeline (from 23.89 ± 0.40 to 26.39 ± 0.45; Figure 2A); CTB-Sap mice showed a slight decrease in their body weight at 7 and 14 dpl, followed by a constant increase up to 49 dpl, but notably CTB-Sap-Mdivi-1 mice had a more pronounced weight loss after lesion and maintained a lower average body weight during the whole experimental period (mixed ANOVA, on groups: *p*-value > 0.05; on time points: *p*-value < 0.001; on groups x time points interaction: *p*-value < 0.001; Figure 2A). Significant differences from pre-lesion levels were observed at 14 dpl (weight loss after lesion; Sidak post hoc test for repeated measures: *p*-value < 0.05; Figure 2A) and at 35–49 dpl (body weight recovery; Sidak post hoc test for repeated measures: *p*-value < 0.05; Figure 2A).

### 2.1. Mdivi-1 Reduced Motor Deficits after CTB-Sap Lesion

A clinical score modified from Albano and colleagues [39] was used to evaluate the functional deficit of the right hindlimb (RHL) by observing a series of signs including alterations of posture, foot placement, toe spread, and ability to produce forward motion. The observations were done weekly: at 7 dpl, all lesioned animals had a severe worsening of RHL functions, which was followed by a partial recovery up to 49 dpl (Friedman test, on CTB-Sap-Mdivi-1 group: *p*-value < 0.05; on CTB-Sap-vehicle: *p*-value < 0.001; Figure 2B). Notably, this functional worsening was significantly lower in CTB-Sap-Mdivi-1 mice, compared to CTB-Sap-vehicle, at both 7 and 14 dpl (Kruskal-Wallis test, *p*-value < 0.05). To analyze sensorimotor coordination, lesioned and CTRL mice were weekly subjected to a grid walk test, and the number of footfalls for each limb was counted (Figure 2C–F). An increase in total footfalls (relative to all limbs) was found at 7 dpl in all lesioned mice, and this was followed by a recovery up to control levels until the end of the experimental period (mixed ANOVA, on time points: *p*-value < 0.001; Figure 2C). However, the total number of errors in all grid walk sessions pooled together was significantly higher in vehicle-treated mice compared to those treated with Mdivi-1 (one-way ANOVA, *p*-value < 0.001; Tukey’s post hoc test, *p* < 0.05; Figure 2E). The difference between CTB-Sap-vehicle and CTB-Sap-Mdivi-1 is greater at 7 dpl (10-fold vs. 6-fold, *p*-value < 0.05; Figure 2C), and the number of footfalls scored by vehicle-treated mice remained significantly higher than CTRL levels until 28 dpl, whereas animal treated with Mdivi-1 showed a faster recovery (Figure 2C). A similar trend was evident from the analysis of RHL errors (Figure 2D,F), showing a 13- and 9-fold increase in footfalls at 7 dpl in vehicle- and Mdivi-1-treated, CTB-Sap-lesioned mice, respectively, followed by a recovery that appeared faster in treated animals.

### 2.2. Effects of CTB-Sap Lesion and Drug Treatment on the Gait Parameters

Control and lesioned mice were allowed to walk on a flat acrylic surface to evaluate a series of gait parameters, including base of support, average position of feet during stance, swing speed, and duration of stance and swing phases [40]. Animals were tested at 7 and 49 dpl, and all parameters were derived by analyzing videos and evaluating feet positions in relation to the body axis over time (Figure 3 and Figure 4). In particular, for each animal, the average positions of feet for all limbs (right forelimb, RFL; left forelimb, LFL; right hindlimb, RHL, and left hindlimb, LHL) were determined and compared across groups (Figure 3A–E). The coordinates of the foot position were reported in relation to the position of the tail base (that represents the origin of the coordinates system) and the same applied to the average position of the neck and body center (red dots in Figure 3A–E). As the mouse body is moving forward during stance, the relative position of each limb during stance is moving backward, so the position of each foot at its initial contact with the surface represents the anterior extreme position (AEP, i.e., the relative foot position at the beginning of the stance phase; blue dots in Figure 3A–E), while the foot position (relative to the body) at the end of the stance phase represents the posterior extreme position (PEP, orange dots in Figure 3A–E). The distance between AEP and PEP for each paw is defined as stance trace length, that is the amount of body wobble during stance phases, and it defines the total anterior and posterior excursion of each limb during stepping. In CTRL mice, this value was ≃1.5 cm for hindlimbs and 2.2 cm for forelimbs (Figure 3A). As seen in Figure 3A–H, the mean value of stance trace length relative to RHL was significantly reduced after lesion, and a small, but significant, recovery was observed in lesioned mice that received the drug vehicle alone (mixed ANOVA, *p*-value < 0.05; Figure 3A–H), whereas the stance traces relative to the other limbs appeared to be unchanged after lesion and drug treatments (mixed ANOVA, *p*-value > 0.05; Figure 3A–H). 

To evaluate the base of support of animals during stance, the distance of hindlimbs from the body axis was measured at the AEP. In CTRL mice, this value was of ≃ 1.0 cm for forelimbs and 1.3 cm for hindlimbs. After CTB-Sap lesion, an increase in RHL distance was found at 7 dpl (mixed ANOVA, *p*-value < 0.05; Figure 3I), which returned to control levels at 49 dpl in both Mdivi-1-treated and vehicle-treated mice (mixed ANOVA, *p*-value > 0.05; Figure 3I). Conversely, no variation of distance was observed for the other limbs (mixed ANOVA, *p*-value > 0.05; Figure 3J).

Another parameter used to evaluate gait modifications after CTB-Sap lesion and/or treatments was the stride length, which is the length of a step, defined as the distance between two consecutive foot positions at touchdown. In CTRL, a stride length of ≃4.7 cm and 4.0 cm was found for hindlimbs and forelimbs, respectively. This parameter appeared to be significantly reduced for RHL in lesioned animals at 7 dpl (mixed ANOVA, *p*-values < 0.05; Figure 4A), but near-normal values were found at 49 dpl in both Mdivi-1-treated and vehicle-treated mice, thus suggesting a spontaneous recovery (mixed ANOVA, *p*-value > 0.05; Figure 4A). Conversely, no variation in stride length was found for the other limbs (mixed ANOVA, *p*-value > 0.05; Figure 4B).

The ratio between swing and stance duration represents another important parameter to evaluate a normal gate. In CTRL animals, a ratio of ≃0.15 and 0.20 was measured for hindlimb and forelimb, respectively, and this parameter was also affected by the CTB-Sap lesion: unexpectedly, no variation was found for RHL (mixed ANOVA, *p*-value > 0.05; Figure 4C), whereas the other limbs showed a significant reduction in this parameter at 7 dpl (mixed ANOVA, *p*-value < 0.05; Figure 4D) followed by a spontaneous recovery at 49 dpl (mixed ANOVA, *p*-value > 0.05; Figure 4D), thus suggesting an increased stance duration and an increased swing speed in the unaffected limbs to compensate for the impaired RHL function.

In fact, the swing speed was also altered after CTB-Sap lesion. In CTRL mice, a swing speed of ≃42 and 33 cm/s was measured for hindlimbs and forelimbs, respectively. After the lesion, at 7 dpl, swing speed appeared reduced for RHL (≃28 cm/s) and increased in LHL (≃55 cm/s) and forelimbs (≃40 cm/s). In particular, the ratio between LHL and RHL swing speed appeared to be increased by ≃70% above CTRL levels in lesioned animals at 7 dpl (mixed ANOVA, *p*-value < 0.05; Figure 4E), with no significant difference between treated and untreated mice, and this parameter (named claudication) appeared to be reduced at 49 dpl and completely normalized in Mdivi-1-treated mice (mixed ANOVA, *p*-value > 0.05; Figure 4E), while it remained significantly higher than CTRL levels in vehicle-treated subjects (mixed ANOVA, *p*-value < 0.05; Figure 4E).

### 2.3. Mdivi-1 Treatment Has Neuroprotective Effects on Motoneurons and Promotes Their Plasticity

The surviving MNs in the lumbar SC of mice receiving CTB-Sap lesion and then treated with either Mdivi-1 or vehicle alone were counted under fluorescence microscopy. The analysis of choline acetyltranferase (ChAT)-positive MN profiles revealed a marked and significant decrease in MN number by 30.8% in the right SC side (lesioned side) compared to the left (untreated) side (Kruskal–Wallis one-way analysis of variance: *p*-value < 0.05; Figure 5A,C,D). Conversely, Mdivi-1-treated mice showed a 11.4% reduction in MN number in the lesioned side (Kruskal–Wallis one-way analysis of variance: *p*-value > 0.05; Figure 5A,C,E), suggesting a significant neuroprotective effect of treatment when comparing the lesioned spinal cord sides of vehicle-treated and Mdivi-1-treated mice (Kruskal–Wallis one-way analysis of variance: *p*-value < 0.05; Figure 5A,D,E).

Moreover, in order to evaluate possible plastic changes in MNs as a result of lesion and/or drug treatment, the MN soma size was measured in the lamina IX of the SC lumbar region innervating the gastrocnemius muscle. It was interesting to observe that spared MNs in both SC sides of lesioned mice treated with Mdivi-1 showed a significant increase in the mean neuronal area compared to the vehicle group (one-way ANOVA: *p*-value < 0.01; Figure 5B), and that this increase was significantly greater in the intact side compared to the lesioned one (Tukey’s post hoc test, *p*-value < 0.05; Figure 5B).

### 2.4. Effects of CTB-Sap Lesion and Drug Treatment on Synaptic Function

To address the question whether the beneficial effects of Mdivi-1 on motor function could be supported by synaptic plasticity, the expression of Synapsin-I has been measured in the lamina IX of lumbar SC, which is the SC region containing the MN pool innervating the CTB-Sap-injected gastrocnemius muscle. Synapsin-I immunoreactivity has been quantified in the lesioned side of the SC and compared to the intact SC. Unexpectedly, a ≃35% reduction in Synapsin-I has been found in all lesioned animals compared to normal tissue (one-way ANOVA: *p*-value < 0.001; Figure 6A, compare Figure 6C with Figure 6E,G), with no effects of the drug treatment (compare Figure 6E,G). Interestingly, an intense Synapsin-I immunoreactivity is evident surrounding the MN membranes, suggesting a large number of synaptic contacts that are likely more abundant in the intact compared to lesioned animals (arrowheads in Figure 6C,E,G).

### 2.5. Muscle Denervation Leads to Atrophy

Loss of MNs and muscular denervation are common hallmarks of CTB-Sap-injured mice, as well as of neurodegenerative diseases such as ALS. To assess muscle pathology, the cross-sectional area of myofibers has been measured in the gastrocnemius muscles of CTB-Sap mice after hematoxylin–eosin staining. The analysis showed signs of atrophy in the injected gastrocnemius muscles of all lesioned mice and the average cross-sectional area of the muscle fibers was ≃30% of the normal area (one-way ANOVA: *p*-value < 0.001; Figure 7A,C). Similarly, a significant weight loss was observed in injected muscles compared to the contralateral side (one-way ANOVA: *p*-value < 0.001; Figure 7B). Interestingly, a large number of centrally located nuclei were found in denervated muscles, thus suggesting an attempt at regeneration (Figure 7C, arrows in left and central panels), and a small but significant increase in the cross-sectional area was observed in the intact (left) muscle of lesioned animals treated with Mdivi-1, as a possible attempt to compensate for the decrease in support in the lesioned side (Tukey’s post hoc test: *p*-value < 0.05; Figure 7A). Muscle denervation after CTB-Sap injection and the consequent MN depletion has been functionally confirmed by EMG, which demonstrated that intact muscles were totally silent in anesthetized mice (Figure 7D), whereas spontaneous activity, i.e., fibrillations (Figure 7E) and positive sharp waves (Figure 7F), were recorded in the injected muscles at both two and six weeks after the lesion (Figure 7E,F), without difference between the time points.

## 3. Discussion

Mitochondrial defects are thought to play a key role in the progression of neurodegenerative diseases, including ALS [33,37,38,41].

A growing body of evidence indicates that excessive mitochondrial fragmentation is caused by aberrant mitochondrial fission mediated by Drp-1 and suggests that its inhibition may be beneficial for MN survival [42,43,44,45,46]. It has been reported that high levels of Drp-1 trigger mitochondrial impairment, which in turn causes oxidative stress, a decrease in ATP production, synaptic damage, and the activation of the apoptosis pathway [44,47,48].

The purpose of the present study was to test the potential neuroprotective effect of Mdivi-1, which is known to block the self-assembly of Drp-1 and then inhibit mitochondrial fission. The drug was administered to our mouse model of MN depletion, where a focal removal of spinal MNs has been induced by injection of CTB-Sap in the gastrocnemius muscle and its retrograde suicide transport [15,22,25]. This simple model of selective MN depletion allows us to focus on the functional and molecular mechanisms of neuroplastic changes upon MN removal, without the complex pathogenetic factors emerging throughout a neurodegenerative process such as ALS. In particular, the present pilot study investigated the effects of CTB-Sap injection and subsequent drug treatment with Mdivi-1 by performing longitudinal behavioral tests to assess changes in gait and other locomotor aspects, as well as the underlying cellular mechanisms.

As expected, a few days after CTB-Sap injection in the right gastrocnemius muscle, all animals started to display an evident decline in the motor activity of the RHL that reached a maximum during the first two weeks after the lesion, as the observation of limb motion during free exploration of an open field revealed frequent curling of toes, loss of support, and foot dragging. Motor deficits were accompanied (and caused by) the partial loss of MN innervating the gastrocnemius muscle and located in the lumbar region of the spinal cord, and the muscle denervation is confirmed by the presence of spontaneous EMG activity in anesthetized mice (i.e., fibrillations and positive sharp waves) that are absent as expected in the normal muscles. This functional decline was followed by a spontaneous partial recovery during the experimental period and, interestingly, Mdivi-1 treatment was capable of reducing the early hindlimb deficit despite the presence of a slightly toxic effect of the drug, as demonstrated by the loss of body weight. In accordance with these results, the grid walk test confirmed the beneficial effects of treatment in the preservation of motor performance, as demonstrated by the lower number of errors made with the affected limb, and a faster recovery of RHL motor ability, although a spontaneous recovery (but slower) was seen also in untreated animals. A similar trend was seen when all limbs were included in the footfalls counting, thus suggesting a positive effect on limb coordination. This evidence suggests that the drug, which was administered intraperitoneally, would exert a systemic effect by acting on multiple sites, probably both inside or outside the central nervous system, and not only on the CTB-Sap-injected muscles and suffering MNs. Therefore, further studies are necessary to address the causative link between the known effect of the drug and the observed beneficial effects.

To further evaluate the effects of lesion and/or drug treatments on motor performance, various gait parameters have been considered. Gait parameters are significant indicators of limb function and motor coordination. The results have shown the significant reduction of the stance trace length and stride length relative to the RHL soon after CTB-Sap lesion (7 dpl) and a significant increase in the RHL distance from the body axis during stance, while the other limbs were not affected. Together, these parameters can measure the gait stability and the anterior-to-posterior excursion of the limb during stepping, so the observed alterations indicate a loss of limb mobility, stability, and support, as well as foot dragging during motion [40]. A partial recovery towards normal levels was observed at 49 dpl, indicating a spontaneous compensatory adaptation over time, without a beneficial effect of Mdivi-1 treatment that, in some cases, apparently worsened the gait parameters, probably due to the observed, though limited, toxicity. Another gait parameter that appeared affected by the lesion is the ratio between swing and stance duration [40]. Unexpectedly, it was unaffected for RHL but reduced for LHL at 7 dpl and returned to control levels at 49 dpl, without significant effects of the drug, thus suggesting an early spontaneous compensatory adaptation of the contralateral limb to support the animal after loss of function of the lesioned limb. Moreover, swing speed, a crucial aspect of gait dynamics [40], was also altered, appeared reduced for RHL and increased for LHL and forelimbs. Interestingly, the left/right ratio of the hindlimb swing speed was increased after lesion, as a sign of claudication, and recovered at the end of the experimental period, with an additional beneficial effect of Mdivi-1.

Taken together, behavioral tests have proven that a spontaneous recovery of motor function is possible after the lesion, despite a permanent though moderate MN removal, and that the treatment with Mdivi-1 may improve this recovery. The beneficial effects of the drug were probably limited to some aspects of the motor activity, such as motor coordination, as clearly suggested by clinical scoring and grid walk test results, whereas gait analysis was not able to efficiently reveal the effects of treatment. It is in fact predictable that a good performance on the grid walk test requires effort in motor coordination and an efficient sensorimotor integration, whereas a simple walking task on a regular surface is likely not particularly challenging, thus making gait analysis not so capable of detecting subtle changes like those occurring in the present neurotoxic model, consisting in a partial and unilateral removal of a confined subpopulation of MNs innervating the gastrocnemius muscle only. However, the analysis of gait parameters was useful to provide a deeper functional characterization of the CTB-Sap model. These functional results are consistent with a previous study employing the same CTB-Sap model, showing an increased mitochondrial fragmentation and Drp-1 expression in the denervated muscle [25], so a beneficial effect of a drug known to inhibit Drp-1 activity could be expected. However, the observed effects of treatment on motor coordination cannot be explained only by its action on the affected muscle, and a detailed mechanistic study of mitochondrial dynamics should include, for instance, the spinal cord, cerebellum, motor cortex, basal ganglia, and also some general aspects of metabolism.

Moreover, it is known that plastic events, such as synaptic plasticity, axonal sprouting, and morphological changes, within the spared MN population can be responsible for compensatory adaptation after the loss of function caused by the neurotoxic removal of a spinal MN subset [22,25,28]. These spontaneous plastic changes are known to take place also in ALS models, but their ability to sustain motor function is transient and incapable of counteracting disease progression [27,28,49]. Therefore, an ideal therapeutic approach should be capable of both improving plastic changes and supporting neuroprotection to slow down MN degeneration.

Our results revealed a significant difference in the number and size of MNs between treated and vehicle mice, indicating that Mdivi-1 has a neuroprotective effect on damaged MNs. It seems that the treatment has promoted the survival and plasticity of the spared MNs, allowing them to adapt in response to the need to compensate for MNs’ degeneration. Similar effects were seen in models of Parkinson’s disease, where the inhibition of Drp-1 can improve dopaminergic neurogenesis and neuroprotection [50,51], and other studies have shown similar effects in different models of neurodegenerative diseases [37,38]. The phenomena of MNs attempting to form new connections and adapt to new conditions in the tissue microenvironment in response to tissue damage or neuronal loss have been well documented in the literature [22,23,25,26,52,53,54,55,56]. This process may result in an increase in soma size and dendritic complexity of surviving MNs, which might be attributable to their active hunt for new synapses and increased synaptic efficacy. Therefore, it is possible that the observed increase in MNs’ size only in Mdivi-1-treated mice is proof of neuronal adaptation, promoted by the known activity of the drug onto mitochondrial dynamics, and likely involving MN itself but also the whole sensorimotor spinal cord circuitry and supraspinal pathways. Interestingly, soma size plasticity in surviving MNs has also been found in ALS animal models [27,49]. In line with previous studies [20,21,22], Synapsin-I expression was reduced after lesion, but, unexpectedly, the observed plastic and neuroprotective effects of Mdivi-1 were not accompanied by a recovery of Synapsin-I levels in the lumbar SC. However, it has to be considered that the levels of Synapsin-I expression are not limited to the synaptic contacts with the MN somata, reflecting instead the synaptic activity of the whole SC ventral horn.

Finally, an evident atrophy was observed in the muscle following denervation caused by CTB-Sap-induced MN death, as affected muscles showed reduced weight and average fiber diameter. Interestingly, mice treated with Mdivi-1 showed a small but significant increase in the average fiber diameter in the contralateral (intact) gastrocnemius muscle, thus confirming gait results and suggesting that the treatment may sustain a systemic compensatory process promoting support and locomotion. Moreover, all lesioned muscles have shown an increase in the amount of centrally located nuclei, which represents a sign of regeneration [57].

## 4. Materials and Methods

### 4.1. Animal Model

All experiments were performed in accordance with the principle of the Basel Declaration as well as to the European Communities Council directive and Italian regulations (EEC Council 2010/63/EU and Italian D.Lgs. no. 26/2014). The study was conducted in accordance with the recommendations of the local committee for animal welfare (OPBA, University of Catania, Via Santa Sofia 97, Catania, Italy), and the protocol was approved by OPBA and by the Italian Ministry of Health (Authorization number: 566/2020-PR).

A total of 26 male mice were used in this study. The animals, 129S1/SvImJ (Jackson Laboratory), had an average weight of 24.6 g and were 8 weeks old. Animals were randomly assigned to different cages (*n*  ≤  5 animals per cage) and kept under constant temperature (23–25 °C), under a 12/12 h light/dark cycle with ad libitum access to food and water. 

In order to induce MN depletion, mice (*n* = 17) were anesthetized with isoflurane (4% induction, 2% maintenance) and received two injections of the retrogradely transported, ribosome-inactivating toxin, CTB-Sap (Advanced Targeting Systems, San Diego, CA, USA) into the medial and lateral right gastrocnemius muscles, respectively, with a toxin dose of 6 μg/2 μL in PBS per injection, as previously described [22]. This neurotoxin is a tracer-toxin formed by the plant-derived toxin saporin (extracted from *Saponaria officinalis*) coupled with the B subunit of cholera toxin, which serves as a targeted carrier [15]. Subgroups of CTB-Sap-lesioned mice received an intraperitoneal injection of either Mdivi-1 (50 mg/kg in DMSO; *n* = 9) or vehicle alone (*n* = 8), 3 days post lesion (dpl). Treatments were repeated once a week at 10, 17, and 24 dpl. In order to confirm that the toxic effects of saporin depend on the specific targeting, as well as on the possibility to be internalized by neurons, a small group of mice (*n* = 4) received an equivalent dose of unconjugated saporin. An additional group of intact and untreated mice (*n* = 5) served as the control.

Behavior tests were performed to evaluate motor coordination, posture, and gait capability. At the end of the experimental period (49 dpl), animals were perfused, and muscles and the SC were isolated and post-fixed, embedded in paraffin or optimal cutting temperature (OCT) medium, respectively, for post mortem analyses.

### 4.2. Behavioral Tests

A clinical scoring method modified from Albano and colleagues [39] was used to weekly evaluate motor performance. Mice were free to move and explore an open field arena, and the score was determined by considering the following criteria: -*score* 0 = healthy;-*score* 1 = during the tail suspension test, collapse or partial collapse of leg extension towards the lateral midline;-*score* 2 = toes curl under at least twice during walking of 30 cm, or any part of the foot is dragging along the table;-*score* 3 = rigid paralysis or minimal joint movement, or foot not being used for generating forward motion.

Global motor coordination, posture, and balance were assessed weekly using a grid walk test. Animals were placed along a 40 cm-long runway of round metal bars (5 mm diameter) placed at about 1.5 cm from each other, and they were filmed using a camera placed in line with the grid to quantify the number of footfalls in each session. The videos were analyzed using KMPlayer software (version number: 2023.6.29.12).

For gait analysis, at 7 and 49 dpl, mice were allowed to walk on a transparent acrylic glass surface, along a 40 cm-long runway (width: 6.5 cm), and videos were recorded from below. The coordinates of foot position during walking were determined by a homemade method, and a series of gait parameters were calculated, including base of support, average position of feet during stance, swing speed, and stance and swing duration [40].

### 4.3. Electromyography

EMG analysis was performed at 14 and 42 dpl using a portable EMG device (TruTrace Traveler EMG, Deymed Diagnostic, Hronov, Czech Republic) with a bipolar concentric needle electrode inserted into the gastrocnemius and a ground electrode positioned on the skin. Animals were anesthetized with isoflurane before the electromyographic recording. The spontaneous electrical activity of the lesioned and contralateral (non-lesioned) muscle was recorded and then analyzed using the TruTrace software (version 7.0; Diagnostic, Hronov, Czech Republic).

### 4.4. Immunofluorescence Staining

For immunofluorescence staining, samples were washed in PBS to remove OCT. Non-specific sites were blocked using a solution of 5% normal donkey serum (NDS) in 0.4% Triton X100 in PBS for 1 h at room temperature (RT) in a humidity chamber. Then slides were incubated with primary antibody added to 2% NDS and 0.3% Triton X100 in PBS overnight a +4 °C in a humidity chamber. The following primary antibodies were used: goat polyclonal anti-choline acetyltranferase (anti-ChAT; Millipore, Merck group, Darmstadt, Germany; Cat#AB144P-200UL, 1:200 dilution) and rabbit polyclonal anti-Synapsin-I (Abcam Limited, Cambridge, UK; EPR23531-50, ab254349, 1:1000 dilution). Slides were then washed in PBS and incubated for 60 min at RT in a humidity chamber with an appropriate secondary antibody diluted 1:500 in 1% NDS in PBS and 0.3% Triton X100: donkey anti-goat, Alexa Fluor 546 (Invitrogen, ThermoFisher Scientific, Waltham, MA, USA; Cat#A-11056, RRID: AB_142628) and donkey anti-rabbit, Alexa Fluor 488 (Invitrogen, ThermoFisher Scientific, Waltham, MA, USA; Cat# A21206; RRID: AB_2535792). After washing in PBS, nuclei were counterstained with DAPI (1:1000, Invitrogen, ThermoFisher Scientific, Waltham, MA, USA) diluted in PBS for 10 min and sections were coverslipped with BrightMount (Abcam Limited, Cambridge, UK; Cat#ab103746). Images were acquired using a fluorescence microscope (Nikon Europe B.V., Amstelveen, The Nederlands) and analyzed using ImageJ software (version number: 1.53e). For quantitative evaluations (number and size of MNs or protein expression), at least 3–5 sections per animal were included in the analysis.

### 4.5. Measurements of Motoneuron Number and Body Size

For MN counting and body size determination, images of ChAT-stained sections were acquired using fluorescence microscopy from the left and right ventral horns of L3–L5 spinal cord segments, using the same magnification (20× objective). Then, all MN profiles clearly showing the nucleolus were counted, and the number of MN profiles in the lesioned spinal cord side were compared to the contralateral and then expressed as percent value of the contralateral (intact) MN number. Then, using ImageJ software, all MNs included in counting were measured for quantifying their cross-sectional area (body size). The average MN body size in the lesioned side was compared to the contralateral and then reported as a percent of the contralateral (intact) value.

### 4.6. Hematoxylin and Eosin Staining

For analysis of muscle fiber morphology, the gastrocnemius muscles were isolated, weighted, and post-fixed in 4% paraformaldehyde at 4 °C overnight. Muscles were dehydrated using an increasing scale of alcohol (ethanol/water mix 50%, 70%, 90%, 100%) and xylene, and then incubated with paraffin (Bio-Optica Milano S.p.A, Milano, Italy) at 60 °C for 1.5 h. Muscles were then embedded in paraffin blocks, and sections were stained with hematoxylin and eosin (HE) for histological analyses. Images of slides were acquired using a ScanScope digital scanner (Aperio, Bristol, UK) and analyzed by using ImageJ software.

### 4.7. Statistical Analysis

Data were analyzed either as raw data or as mean ± SEM (standard error of the mean), as appropriate. Differences between experimental groups were evaluated by using one-way or two-way ANOVA, followed by Bonferroni’s post hoc test, or using non-parametric test (Kruskal–Wallis or Friedman tests) where appropriate. For all experiments, a *p*-value of < 0.05 was considered to be significant. All analyses were performed by means of Systat package version 11 (Systat Software, Inc., Chicago, IL, USA).

## 5. Conclusions

The results of the present study have confirmed that the CTB-Sap model is a valid tool for research in MN diseases, proving that compensatory plastic changes may take place after the removal of a spinal MN subset. Moreover, it seems likely that treating this animal model with a drug known to inhibit mitochondrial fission may increase this intrinsic plastic capability and protect MNs from degeneration. In conclusion, this pilot study supports the possibility that developing agents that target the homeostasis of mitochondrial dynamics could have therapeutic value in alleviating neurodegenerative disease. Further studies using the CTB-sap model and also a murine ALS model are ongoing to fully understand this intriguing aspect of neurodegeneration and also by a mechanistic evaluation of the functional link between mitochondrial dynamics and functional effects

## Figures and Tables

**Figure 1 ijms-25-07059-f001:**
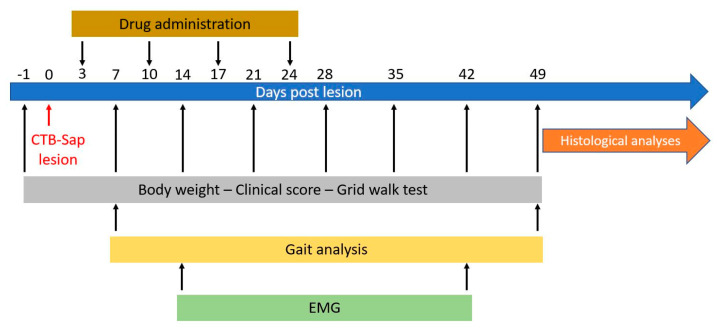
Schematic representation of the experimental timeline.

**Figure 2 ijms-25-07059-f002:**
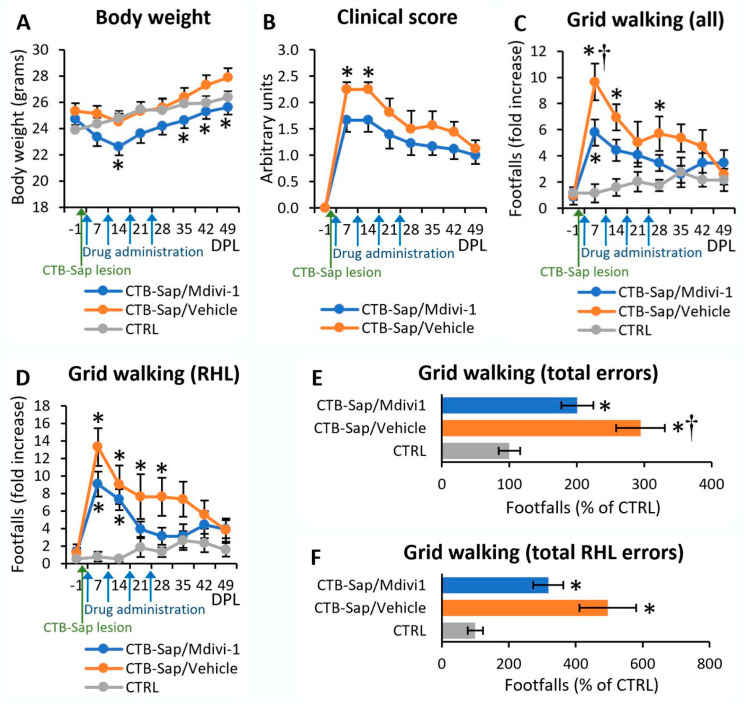
Functional analyses of control and lesioned animals. (**A**) Measurements of the average body weight before and after CTB-Sap lesion and drug administration, in comparison to control (CTRL) animals; asterisks indicate significant differences from pre-lesion levels. (**B**) Clinical score measurements of CTB-Sap mice with or without the administration of Mdivi-1; asterisks indicate significant differences between groups. (**C**,**D**) Number of footfalls relative to all limbs (**C**) or to the right hindlimb (RHL) only (**D**) during the weekly execution of the grid walk test; values are expressed as fold increase from pre-lesion levels. In (**E**,**F**), the number of footfalls counted during grid walk test at all time points are pooled together and expressed in % of the CTRL values to compare the groups in relation to the total number of errors relative to all limbs (**E**) or the RHL only (**F**). DPL: days post-lesion. In all graphs, asterisks indicate significant difference from CTRL animals while (†) indicates significant difference between vehicle-treated and Mdivi-1-treated CTB-Sap animals.

**Figure 3 ijms-25-07059-f003:**
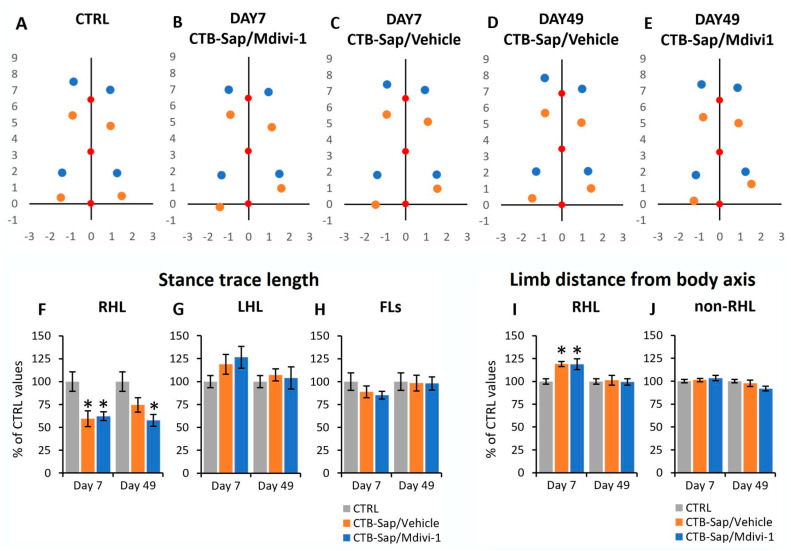
Functional analyses of control and lesioned animals by gait analysis. (**A**–**E**) Average position of feet during stance in relation to the body axis in control (CTRL) and lesioned animals (with or without Mdivi-1 treatment) at 7 and 49 days post-lesion; coordinates refer to the average position of tail base (lower red dot), representing the origin of the coordinates system; the upper red dot represents the average position of the neck, while the middle one represents the center of the body; blue dots indicate the average positions of feet for the forelimbs (upper ones) and hindlimbs (lower ones) at the beginning of stance phase (AEP: anterior extreme position), while orange dots indicate the position of feet at the end of stance phase (PEP: posterior extreme position); for each limb, the distance between AEP and PEP is indicated as stance trace length; coordinates are expressed in cm. (**F**–**H**) Mean values of stance trace length relative to right hindlimb (RHL), left hindlimb (LHL), and both forelimbs (FLs) together. (**I**,**J**) Average limb distance from the body axis, relative to RHL or the other limbs together (non-RHL). In all graphs, asterisks indicate significant differences from CTRL values. All values are normalized to control levels and expressed as mean ± SEM; actual values can be found in the main text.

**Figure 4 ijms-25-07059-f004:**
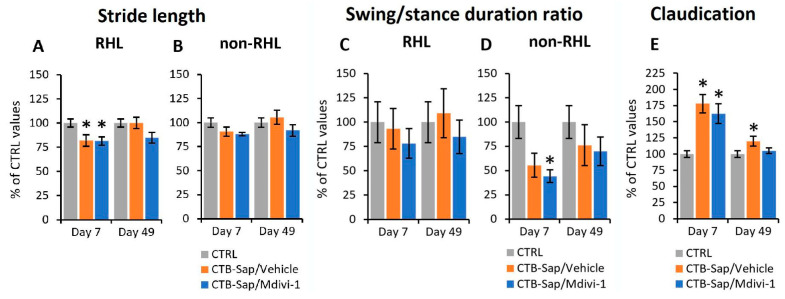
Functional analyses of control and lesioned animals by gait analysis. (**A**,**B**) Mean values of the stride length (distance between two consecutive foot positions at the beginning of the stance phase) relative to the right hindlimb (RHL) or the other limbs together (non-RHL). (**C**,**D**) Mean values of the ratio between swing and stance duration for RHL and non-RHL. (**E**) Average values of the parameter named claudication, which is the ratio between the LHL and RHL swing speed. In all graphs, asterisks indicate significant differences from CTRL values. All values are normalized to control levels and expressed as mean ± SEM; actual values can be found in the main text.

**Figure 5 ijms-25-07059-f005:**
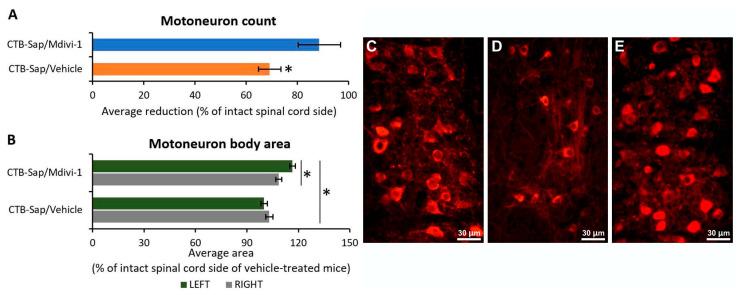
Graphic illustration of motoneuron numbers (**A**) and their soma size (**B**) in the lumbar spinal cord of CTB-Sap-lesioned mice treated with either Mdivi-1 or vehicle alone. (**C**–**E**) Representative images taken with the fluorescence microscope, showing ChAT-positive MN profiles from a normal (**C**) spinal cord (contralateral to the lesion side) or from lesioned spinal cord side of vehicle-treated (**D**) and Mdivi-1-treated mice (**E**). The asterisks indicate significant difference (*p*-value < 0.05).

**Figure 6 ijms-25-07059-f006:**
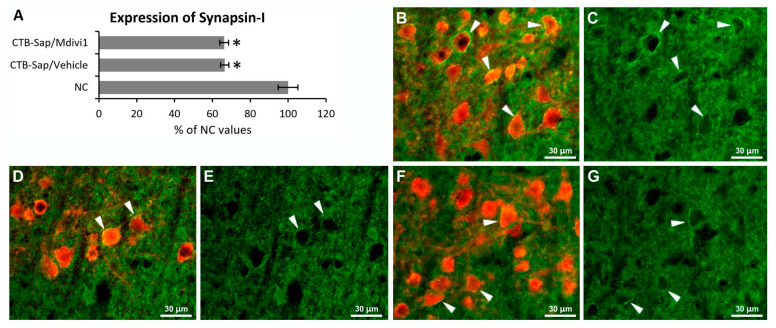
Quantification of Synapsin-I expression by optical density measurement (**A**) in the lamina IX of the lumbar spinal region containing the motoneurons innervating the gastrocnemius muscle. (**B**–**G**) Representative fluorescence images showing the expression of Synapsin-I (green) surrounding the ChAT-positive MN profiles (red) in normal (**B**,**C**), lesioned and vehicle-treated (**D**,**E**) or Mdivi-1-treated (**F**,**G**) spinal cord. Arrowheads indicate synaptic contacts with the motoneuronal membrane. The asterisks indicate significant difference (*p*-value < 0.05).

**Figure 7 ijms-25-07059-f007:**
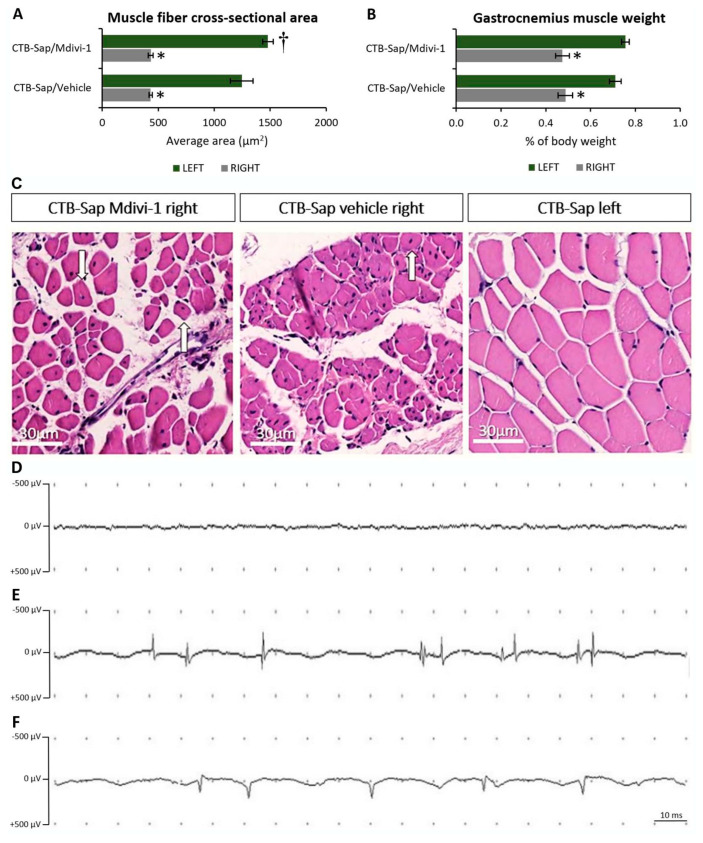
Graphic illustration of muscle atrophy and denervation after CTB-Sap lesion. The graphs represent the measurement of muscle fiber cross-sectional area (**A**) and the muscle weight (**B**) of the right (CTB-Sap-injected) and left (intact) gastrocnemius muscles, after treatment with either Mdivi-1 or vehicle alone; asterisks (*) indicate significant difference from contralateral side, while (†) indicates significant difference from vehicle-treated mice. (**C**) Representative images of hematoxylin–eosin stained sections showing the evident reduction of fiber diameter after lesion, compared to the intact side (right panel); arrows indicate centrally located nuclei in muscle fibers of lesioned animals. (**D**–**F**) Examples of EMG signals showing the absence of spontaneous muscle activity in normal muscles of anesthetized mice (**D**) compared to the presence of fibrillations (**E**) and positive sharp waves (**F**) at six weeks after CTB-Sap injection.

## Data Availability

The data presented in this study are available on request from the corresponding author.

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
