# Peer review of "Mitigating the Functional Deficit after Neurotoxic Motoneuronal Loss by an Inhibitor of Mitochondrial Fission"

_ijms, 2024, doi:10.3390/ijms25137059_

Round 1

Reviewer 1 Report

Comments and Suggestions for Authors

In this manuscript, Ciuro and colleagues investigate the potential effect of Mdivi-1 administration, a specific inhibitor of the mitochondrial fission, for the recovery of motor functions in a murine model of ALS achieved by CTB-Sap injection. The results here presented show the effect of Mdivi-1 for specific motor functions and histological parameters. Overall, the work presents many limitations in the experimental plan, the technical approach and the results organization. As detailed below, these limitations make very difficult the interpretation of the data and not properly support the conclusion. 

Major concerns:

1) The assumption of the work is that Mdivi-1 exerts a neuroprotective effect in pathologies with an abnormal mitochondrial fragmentation and/or dysregulation of Drp1 expression, such as Alzheimer’s and Parkinson’s disease, and ALS. However, it is peculiar that in the presented manuscript the authors did not investigate the mitochondrial dynamic in the CTB-Sap model after the exposure to Mdivi-1. In this regard, the title and many of the conclusions are misleading since a direct link between the recovery of motor functions and the inhibition of mitochondrial fission is not demonstrated. In my opinion, an initial characterization of specific mitochondrial parameters, as the authors did in the previous cited work of Vicario et al., is necessary

2) If I have correctly understood, the administration of Mdivi-1 in lesioned mice promoted only a significant variation in the grid walking performances at day 7 and in the total error, in comparison with lesioned mice treated with the vehicle (as the symbol † seems to indicate). At the same time, Mdivi-1 administration failed the recovery of other parameters as those shown in other Figure 1 graphs, and in the entire Figure 2 and 3, being the statistic referred only to the comparison with healthy control. Including the negative results is always commendable, but the authors have to consider that the behavioral results here showed are excessively limited and risky for supporting the conclusion that Mdivi-1 is able to promote a partial functional recovery in mice.

3) Figure 4A presents two issues. First, authors only show the histograms relative to the quantification of motoneurons. Being the experiment based on fluorescence microscopy aimed at the analysis of ChaT-positive staining, the authors must show the relative representative microscopy images including control, vehicle- and Mdivid-1-treated samples. Second, the statistical analysis is incomplete. Although the data are represented as a function of control (in this case, as the percentage of motoneurons in the intact side of spinal cord), authors should analyze whether the reduction caused by lesion is significant, and only then the statistical significance of the difference between vehicle- and Mdivid-1-treated samples.

4) The method relative to the determination of the data in Figure 4B is not clear and, apparently, is not made explicit in the Materials & Methods section. In my opinion, authors should dedicate to it a specific subparagraph. Anyway, if the method is based again on microscopy, they must show representative images as for the previous point.

5) It is not clear what the authors show in Figure 5B and C since they do not report any comments in the text. It appears that the images are referred to the control and show how the quantification in Figure 5A was made. As already discussed, representative microscopy images are mandatory for each experiment.

6) Data in Figure 6D-F appears incomplete. First, the figure E and F show EMG signals of lesioned mice at two different time points; thus, for a correct comparison of the data, two EMG signals of control mice at matching time points are needed. Second, EMG data appears out of the scope of the manuscript since authors only show EMG data for CTB-Sap mice. Considering the manuscript aim, a comparison between EMG traces taken from CTB-Sap treated with the vehicle and Mdivi-1 must be included.

7) In the statistical evaluation, the authors often report a p value of 0.00. Technically, the p value is never absolute 0 albeit it comes out from the statistical software (it depends on the settings). However, even when the p value tends to 0, it is more appropriate indicate it as p<0.001. 

Minor comments:

The timeline used for the experiments (molecules administration, timing and behavioral tests) is not immediately clear. To understand the rationale, the readers should go as far as the Material and Method section. I would suggest to prepare an additional figure to add in the panel of Figure 1 showing a timeline with the experiment rationale. 

Authors should pay attention in the use of acronyms. For example, the acronym RHL is used already at page 2 but it defined only at page 3. 

Comments on the Quality of English Language

Only minor English editing is required

Author Response

In this manuscript, Ciuro and colleagues investigate the potential effect of Mdivi-1 administration, a specific inhibitor of the mitochondrial fission, for the recovery of motor functions in a murine model of ALS achieved by CTB-Sap injection. The results here presented show the effect of Mdivi-1 for specific motor functions and histological parameters. Overall, the work presents many limitations in the experimental plan, the technical approach and the results organization. As detailed below, these limitations make very difficult the interpretation of the data and not properly support the conclusion. 

Major concerns:

1) The assumption of the work is that Mdivi-1 exerts a neuroprotective effect in pathologies with an abnormal mitochondrial fragmentation and/or dysregulation of Drp1 expression, such as Alzheimer’s and Parkinson’s disease, and ALS. However, it is peculiar that in the presented manuscript the authors did not investigate the mitochondrial dynamic in the CTB-Sap model after the exposure to Mdivi-1. In this regard, the title and many of the conclusions are misleading since a direct link between the recovery of motor functions and the inhibition of mitochondrial fission is not demonstrated. In my opinion, an initial characterization of specific mitochondrial parameters, as the authors did in the previous cited work of Vicario et al., is necessary

RESPONSE: the concern expressed by the Referee is important and requires a detailed explanation. Prior to submission, the authors have considered the opportunity to make an analysis of dynamin-related protein 1 (Drp1) expression and/or mitochondrial length, as done in the previous work by Vicario et al. However, after discussion, the authors decided that the inclusion of these analyses would not be fundamental for the following reasons:

1) A characterization of Drp1 expression and mitochondrial length has already been done in the CTB-Sap model (Vicario et al., 2021), showing an increased mitochondrial fragmentation together with up-regulated Drp1 in the lesioned animals, and also suggesting that both conditions might be partially restored by a treatment with clobetasol.

2) Given the already known increased mitochondrial fission in the CTB-Sap model, the present work has focused on the possible use of this mechanism as a pharmacological target. As the drug Mdivi-1 (mitochondrial division inhibitor-1) is known to act as an inhibitor of mitochondrial fission by blocking the Drp1 activity, the authors have supposed that investigating mitochondrial fission would have merely confirmed the already known mechanism of action of the drug. Moreover, quantifying Drp1 expression would have probably provided confusing results, because it is unknown what kind of effect would be exerted on Drp1 expression by blocking its function and, conversely, an increased or decreased Drp1 expression would provide controversial conclusions in presence of a drug acting as an inhibitor of Drp1 activity itself.

3) The reviewer considers that title and conclusions are misleading since a direct link between the recovery of motor functions and the inhibition of mitochondrial fission is not demonstrated, and he/she proposed that an investigation of mitochondrial dynamics (probably by mitotracker) would be useful to address this problem. The authors agree with these considerations and the text (including the title and conclusions) has now been modified in order to avoid any misleading statement, where possible. Concerning the mitochondrial dynamics, although a quantification of mitochondrial fission is possible, in our opinion, it would not provide any direct link with functional results, but just a confirmation of the well-known mechanism of action of Mdivi-1.

4) However, if the above reasons will still not sound convincing to the reviewer, further analyses can certainly be carried out but, in such case, additional time may be necessary.

2) If I have correctly understood, the administration of Mdivi-1 in lesioned mice promoted only a significant variation in the grid walking performances at day 7 and in the total error, in comparison with lesioned mice treated with the vehicle (as the symbol † seems to indicate). At the same time, Mdivi-1 administration failed the recovery of other parameters as those shown in other Figure 1 graphs, and in the entire Figure 2 and 3, being the statistic referred only to the comparison with healthy control. Including the negative results is always commendable, but the authors have to consider that the behavioral results here showed are excessively limited and risky for supporting the conclusion that Mdivi-1 is able to promote a partial functional recovery in mice.

RESPONSE: the reviewer correctly noted that beneficial functional effects was seen in some but not all motor tests. However, more precise information is necessary to address this important comment. First, Fig. 1B shows that a beneficial effect can also be seen by clinical score, which is a modified version of a scoring method used for evaluating ALS animal models (Albano et al., 2013). In particular, Mdivi-1 treatment seems to reduce the functional worsening in the earlier time-points (7 and 14 days), probably as a result of neuroprotective effects, and this assumption is supported by the reduced motoneuronal depletion in Mdivi1-treated mice. In accordance to clinical score data, Figure 1C shows an early effect of the drug (at 7 days) compared to untreated mice and, additionally, it shows that the functional recovery was faster in treated mice, and this was also evident in RHL errors (Fig. 1D). On the other hand, gait analysis only shows a small effect of the drug, which is limited to the parameter named “claudication” (i.e. the ratio between left and right hindlimb swing speed), that is altered in vehicle-treated mice (significantly different from normal) and normalized in Mdivi1-treated animals. This (partial) negative finding has been commented in discussion, and it was not surprising. It should be noted that the lesion is confined to a small motoneuron population innervating the right gastrocnemius muscle, so the analysis of gait performance through a regular surface may likely be not enough to visualise motor deficits. Conversely, grid walk test represents a more challenging test requiring an intact four-limb coordination. The authors decided to include gait analysis, despite the small effect of the drug onto these parameters, as a useful method for deeper functional characterization of the CTB-Sap model. The authors have partially modified the Discussion in attempt to improve the interpretation of data and avoid misleading conclusions.

3) Figure 4A presents two issues. First, authors only show the histograms relative to the quantification of motoneurons. Being the experiment based on fluorescence microscopy aimed at the analysis of ChaT-positive staining, the authors must show the relative representative microscopy images including control, vehicle- and Mdivid-1-treated samples. Second, the statistical analysis is incomplete. Although the data are represented as a function of control (in this case, as the percentage of motoneurons in the intact side of spinal cord), authors should analyze whether the reduction caused by lesion is significant, and only then the statistical significance of the difference between vehicle- and Mdivid-1-treated samples.

RESPONSE: concerning this comment, it should be noted that CTB-Sap has been injected unilaterally in the right gastrocnemius muscle and, therefore, motoneuron depletion was confined to the ipsilateral side of the spinal cord, being the contralateral side absolutely intact. Since no difference is expected between the left spinal side of lesioned animals and the normal spinal cord (as shown in our previous papers), the left side represents an internal control. Therefore, including data from control spinal cord tissue, as well as including a statistical comparison between lesioned and control animals may be redundant, and it has been avoided to save space and simplify the reading of data. The statistical evaluation of motoneuron depletion induced by lesion has already been done in both vehicle and treated groups, but the authors have mistakenly omitted the p-value relative to the treated group, that is now reported in results. Representative images taken from ChAT-stained sections of treated and untreated, lesioned spinal cords have now been included in Figure 4.

4) The method relative to the determination of the data in Figure 4B is not clear and, apparently, is not made explicit in the Materials & Methods section. In my opinion, authors should dedicate to it a specific subparagraph. Anyway, if the method is based again on microscopy, they must show representative images as for the previous point.

RESPONSE: methods have been modified by adding a specific paragraph describing the determination of motoneuron number and body area more clearly. In addition, micrographs showing ChAT-positive motoneurons have been included in Figure 4.

5) It is not clear what the authors show in Figure 5B and C since they do not report any comments in the text. It appears that the images are referred to the control and show how the quantification in Figure 5A was made. As already discussed, representative microscopy images are mandatory for each experiment.

RESPONSE: the content of panels B and C in Figure 5 was described in the figure caption but, as noted by the Reviewer, their citation in the text was mistakenly missing. Now, Figure 5 has been modified by including additional representative images as requested, and its description in the text and figure caption has been updated.

6) Data in Figure 6D-F appears incomplete. First, the figure E and F show EMG signals of lesioned mice at two different time points; thus, for a correct comparison of the data, two EMG signals of control mice at matching time points are needed. Second, EMG data appears out of the scope of the manuscript since authors only show EMG data for CTB-Sap mice. Considering the manuscript aim, a comparison between EMG traces taken from CTB-Sap treated with the vehicle and Mdivi-1 must be included.

RESPONSE: the Results paragraph describing EMG results was probably misleading and it has been revised, together with the figure caption. In particular, panels D, E and F of the Figure 6 show EMG signals (fibrillations in E and positive sharp waves in F) at the same time-point (6 weeks), since no apparent difference was observed between time-points or after treatment. However, a quantitative evaluation of denervation and possible reinnervation would require more complex EMG experiments, with longer data recordings and event counting, that overcome the aim of the present work. So, EMG signals in this figure have just the purpose of showing the occurrence of fibrillations and positive sharp waves (being both signs of denervation and also present in ALS patients) in lesioned animals, compared to the completely silent condition observed, as expected, in the normal muscles. These considerations have now been included in Discussion.

7) In the statistical evaluation, the authors often report a p value of 0.00. Technically, the p value is never absolute 0 albeit it comes out from the statistical software (it depends on the settings). However, even when the p value tends to 0, it is more appropriate indicate it as p<0.001. 

RESPONSE: all those p-values have been corrected as suggested.

Minor comments:

The timeline used for the experiments (molecules administration, timing and behavioral tests) is not immediately clear. To understand the rationale, the readers should go as far as the Material and Method section. I would suggest to prepare an additional figure to add in the panel of Figure 1 showing a timeline with the experiment rationale. 

RESPONSE: the additional figure has been included as suggested.

Authors should pay attention in the use of acronyms. For example, the acronym RHL is used already at page 2 but it defined only at page 3. 

RESPONSE: a careful check of the text has been done as suggested, to correct such mistakes and other typos.

Comments on the Quality of English Language

Only minor English editing is required. 

RESPONSE: a careful revision of English editing has been done prior to re-submission.

Reviewer 2 Report

Comments and Suggestions for Authors The manuscript entitled Functional recovery after neurotoxic motoneuronal loss is pro-moted by the inhibition of mitochondrial fission is scientifically well defined.
It can be considered for publication after minor revision.

1. Introduction section can be improve.   

2. Define the abbreviation at initial look (define MNs).

3.  Magnification of Images must be incorporated in the main text. 

Author Response

The manuscript entitled Functional recovery after neurotoxic motoneuronal loss is promoted by the inhibition of mitochondrial fission is scientifically well defined.

It can be considered for publication after minor revision.

  1. Introduction section can be improve.

RESPONSE: the authors tried to improve the quality of text before re-submission.

  1. Define the abbreviation atinitial look (define MNs).

RESPONSE: a careful check of acronyms and their definition has been done.

  1. Magnification of Images must be incorporated in the main text. 

RESPONSE: the magnification used during image acquisition is now stated in methods. In figures, magnification can be determined by using the scale bars.

Round 2

Reviewer 1 Report

Comments and Suggestions for Authors

Although I appreciate the authors’ efforts in improving the manuscript and replying to my concerns, I still have the opinion that the work, at this stage, is not suitable for publication. This is exclusively due to the limitations in the experimental approach and plan, and to the unconvincing results about the functional recover of mice, as I have highlighted in the previous round of revision. Indeed, most of my concerns remains also after having read the revised version of the manuscript. 

First, the present work does not include any data regarding mitochondrial fragmentation and/or dynamic upon Mdivi-1 exposure in this specific ALS model, nor any functional data (i.e., oxygen consumption, quantification of biogenesis/mitophagy markers by Western blot or Real Time PCR) that may explain, even indirectly, that the slight functional recovery observed in treated mice by the authors depends on a partial recovery of the mitochondrial function. Being Mdivi-1 a drug acting on mitochondria, I would expect that authors investigate this point as a first. 

Second, I do not question the importance of the functional results here displayed, especially considering the aggressiveness of ALS and the absence of any valid pharmacological treatments. However, as I already noticed in the previous round of the revision, the results appear very limited to specific parameters, and without any additional experiment it is risky talking about a “functional recovery”, as state also in the title. 

Comments on the Quality of English Language

Quality of English is acceptable

Author Response

Reviewer 1, round 2.

Although I appreciate the authors’ efforts in improving the manuscript and replying to my concerns, I still have the opinion that the work, at this stage, is not suitable for publication. This is exclusively due to the limitations in the experimental approach and plan, and to the unconvincing results about the functional recover of mice, as I have highlighted in the previous round of revision. Indeed, most of my concerns remains also after having read the revised version of the manuscript. 

RESPONSE: in the previous round of revision, the Authors tried to explicate the evidences, and also the limits of the whole dataset. Moreover, following the suggestions previously provided by the Reviewer, the presentation of results and the description of the experimental plan have been improved (by adding the timeline of experiments and new images in figures, and also by clarifying some aspects of data presentation), but the Reviewer opinion about research design and presentation of results is now the same as round 1: “Must be improved”. The Authors tried to clarify that the most convincing functional evidence is the reduced functional worsening at the earliest time-points (evident and statistically significant at 7 and 14 days by clinical score and at 7 days by grid walking) in the Mdivi1-treated mice. Concerning the functional recovery, the Authors tried to clarify what seems to be evident in figure 2 (previously fig. 1) by grid walking results. In particular, those data clearly shown that although a functional recovery is also present in untreated mice (spontaneous recovery in rodents has already been shown in previous papers), it was faster in treated mice, and this evidence is supported by statistics. The Authors also tried to better justify the (partially) negative results concerning gait analysis, but they are still certain that including these results is useful as a deep characterization of the CTB-Sap lesion model. Given that the Reviewer remains doubtful about functional recovery, the Authors may only try to improve Discussion and Conclusions, as in vivo experiments cannot be repeated.

First, the present work does not include any data regarding mitochondrial fragmentation and/or dynamic upon Mdivi-1 exposure in this specific ALS model, nor any functional data (i.e., oxygen consumption, quantification of biogenesis/mitophagy markers by Western blot or Real Time PCR) that may explain, even indirectly, that the slight functional recovery observed in treated mice by the authors depends on a partial recovery of the mitochondrial function. Being Mdivi-1 a drug acting on mitochondria, I would expect that authors investigate this point as a first. 

RESPONSE: the Authors would try again to discuss their opinion about this point. The reasons previously provided by the Authors remain valid (please see the previous rebuttal letter), but further considerations emerged after discussion among them. Although, from a theoretical point-of-view, the Reviewer’s request (that is, …being Mdivi-1 a drug inhibiting mitochondrial fission, the investigation of the expected effects on mitochondria is mandatory) is reasonable, the experimental accomplishment of this logical sentence has a couple of serious concerns: 1) even if a qualitative and/or quantitative evaluation of mitochondrial fission (that is the target of Mdivi-1) is done and demonstrates an effect of the drug, it would not be a proof of a cause-effect link with functional effects, but just a proof of the already known mechanism of action of the drug. In order to logically demonstrating a direct functional link, additional in vivo experiments are necessary, including for instance a group of lesioned mice where the activity of Mdivi-1 is blocked by a competitor, and a group where mitochondrial fission (and/or fusion) is modulated by a different drug. These studies are beyond the limits of the present pilot study, but they would be part of a future follow-up research project. 2) Measuring other mitochondrial parameters (e.g. oxygen consumption or other metabolic parameters, as suggested) would be interesting and useful to provide additional mechanistic insights, but not necessarily linked to the direct Mdivi-1 action to mitochondria. Also, these experiments would require additional in vivo research activity (not possible), since at the moment only fixed tissues are available, so Western blot or PCR analyses are not possible. 3) Given that the neurotoxic lesion was focal (confined to the motoneuron pool innervating the right gastrocnemius) but both the administration of Mdivi-1 and the main functional results were systemic, being an amelioration of motor coordination (that is requested for a good grid walking performance), it appears difficult to decide where (in which organs) to quantify those parameters. In muscles? In this case it will be necessary to investigate both normal, lesioned and contralateral muscles (please note the significant increase of fiber diameter in the contralateral, intact, muscle), but these data would probably be not useful to support an improved motor coordination. So, it would be necessary to analyze mitochondria in the spinal cord, and/or in motor cortex and/or in cerebellum, but which neuronal (and/or glial) cell population should authors focus on? These aspects are now better discussed in the text (please see Discussion and Conclusions), and the need for future studies including a deep evaluation of the Mdivi-1 mechanism of action is also underlined, although it appears to be beyond the limits of the present pilot study..

Second, I do not question the importance of the functional results here displayed, especially considering the aggressiveness of ALS and the absence of any valid pharmacological treatments. However, as I already noticed in the previous round of the revision, the results appear very limited to specific parameters, and without any additional experiment it is risky talking about a “functional recovery”, as state also in the title

RESPONSE: as discussed above, Authors do not completely agree with the assumption that the beneficial functional effects of the drug are “very limited”. In any case, additional data aiming at the consolidation of functional results would require further in vivo experiments that are not possible, because these experiments would require a new Government authorization and months of research activity.

In conclusion, the Authors would like to make it evident that this study is just a pilot study providing a proof-of-concept showing that an already known inhibitor of mitochondrial fission is likely able to exert beneficial effects (especially at earlier time-points) onto some aspects of motor activity (i.e. motor coordination). The dissection of the mechanisms (mitochondrial dynamics, metabolism, oxidative stress, apoptosis and so on…) underlying these functional effects, require a lot of additional experiments that, in our opinion, overcome the limits of the present study and require a separate mechanistic research plan.

An additional revision of the text, including the title, has been done in the present, second round of revision, to better clarify the above considerations, and the Authors hope that their reasons could be enough and that the manuscript would now be suitable for publication in the present form.
